# Achieving efficient power generation by designing bioinspired and multi-layered interfacial evaporator

Zhuangzhi Sun [1,2] ✉, Chuanlong Han [1], Shouwei Gao[3], Zhaoxin Li [1] ✉, Mingxing Jing [1], Haipeng Yu [2] ✉ & Zuankai Wang [3] ✉

Water evaporation is a natural phase change phenomenon occurring any time and everywhere. Enormous efforts have been made to harvest energy from this ubiquitous process by leveraging on the interaction between water and materials with tailored structural, chemical and thermal properties. Here, we develop a multi-layered interfacial evaporation-driven nanogenerator (IENG) that further amplifies the interaction by introducing additional bionic light-trapping structure for efficient light to heat and electric generation on the top and middle of the device. Notable, we also rationally design the bottom layer for sufficient water transport and storage. We demonstrate the IENG performs a spectacular continuous power output as high as 11.8 µW cm$^{-2}$ under optimal conditions, more than 6.8 times higher than the currently reported average value. We hope this work can provide a new bionic strategy using multiple natural energy sources for effective power generation.

Water evaporation is a ubiquitous physical process that plays an essential role in the global water cycle[1–3]. Less apparently, such a dynamic mass and heat transport phenomenon is also associated with vast energy flow. The first evaporation-based energy harvesting strategy was reported in 2017, which can generate continuous and considerable electricity through water flow within carbon black sheets[4]. Over the past several years, extensive efforts have been done to boost devices' performance[5–7].

Recently, materials with structural, chemical and thermal properties are deeply delved to enhance the efficiency of the energy harvesting. Among them, localized temperature enhancement strategy is the most important to determine the speed of interfacial evaporation on a water surface[8,9]. Solar-heat-driven interfacial evaporation has been identified as a promising green and sustainable solution for the pressing global problem of water shortage which can directly transfer the light to heat for evaporation[10–13]. With an elegant choice of materials, conditions, and structures, the evaporation rate can reach over 4 kg m$^{-2}$ h$^{-1}$ under 1 sun[14–16]. The light absorption efficiency of the surface is the

fundamental bottleneck that restrains further increase in the evaporation performance. Coupled bioinspired strategy[17–21], multi-layered design is considered as an effective measure to alleviate this dilemma.

Here, we developed a simple and efficient interfacial evaporation-driven nanogenerator (IENG) that achieved layered functionalization and introduced bionic light-trapping structure for light to heat and electric generation. Wherein the bottom is made of porous ionic hydrogel for water supply, the middle layer possesses multi-walled carbon nanotube (MWNT), MXene for higher electric conductivity, and the top nanofibers layer composes MWNT for heat and electricity generation. Most importantly, our design harnesses the light-trapping structure on the surface of a moth's eye that exhibits nearly zero reflection of sunlight, and contributes to a high light absorption efficiency of 96.7% and an excellent water evaporation rate of 2.78 kg m$^{-2}$ h$^{-1}$ under a light intensity of 1 sun. Furthermore, resulting from the high evaporation rate, our IENG exhibits an output power density of 11.8 µW cm$^{-2}$ under the modified condition, more than 6.8 times higher than the currently reported average value. Our device demonstrates a

[1]Province Key Laboratory of Forestry Intelligent Equipment Engineering, College of Mechanical and Electrical Engineering, Northeast Forestry University, Harbin 150000, People's Republic of China. [2]Key Laboratory of Biobased Material Science & Technology, Ministry of Education, Northeast Forestry University, Harbin 150000, People's Republic of China. [3]Department of Mechanical and Biomedical Engineering, City University of Hong Kong, Hong Kong, People's Republic of China. ✉e-mail: sunzhuangzhi@nefu.edu.cn; 2014211213@nefu.edu.cn; yuhaipeng20000@nefu.edu.cn; zuanwang@cityu.edu.hk

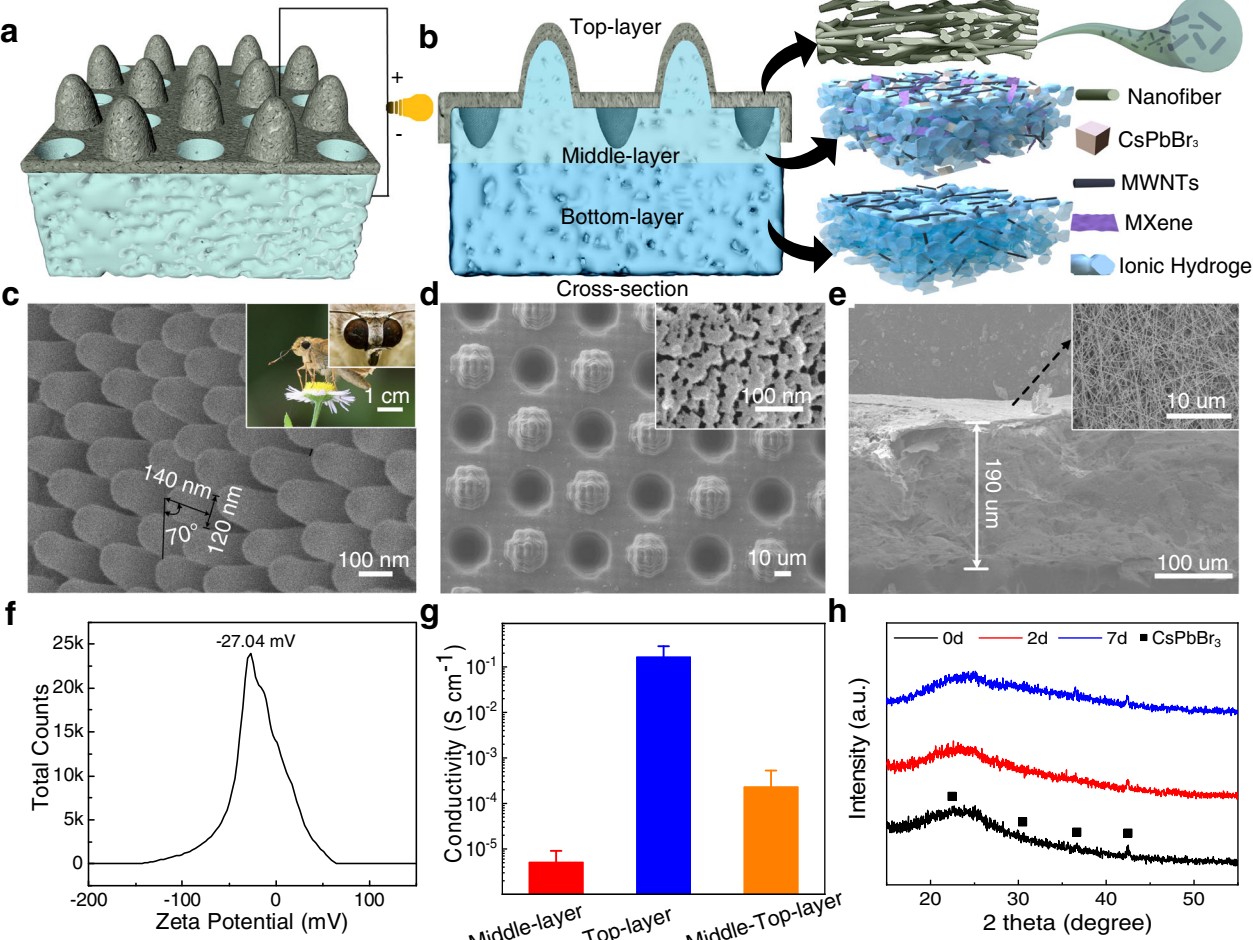

**Fig. 1 | Design and characterization of the IENG. a** Schematic diagram of structural of the IENG for all-in-one evaporator towards simultaneous water evaporation and electricity generation. **b** Schematic diagram of the composition of a fabricated IENG. **c** Morphological characteristics of the moth eye. **d** Surface microstructure of the bionic middle layer. **e** Cross-section SEM image of the top layer. **f** Zeta potential of the IENG. **g** Electrical conductivity of different layers. **h** X-ray diffraction pattern of CsPbBr$_3$ under different evaporation time. The error bars represent standard deviations.

new concept for developing natural interfacial evaporation-driven power generation systems and acts as a ground-breaking attempt to obtain energy harvested from multiple sources in external environments.

## Results

### Design and characterization of the IENG

Figure 1a, b show the design of the IENG. Overall, our IENG is a hierarchical structure possessing three functional layers. The top layer is covered by nanofibers composing MWNT, providing the primary light to heat and electric generation properties. The middle layer is a regular array with a carefully designed size mimicking the moth-eye and composed IH, MWNT, MXene, and CsPbBr$_3$ type perovskite. This bionic light-trapping (BL) structure worked as an accessory to strengthen the light absorption and electric output. The bottom layer comprises ionic hydrogel (IH), which is employed a stable water storage/ supply during evaporation.

The BL structure of the IENG is inspired by bean hawk moth eyes, which consists of hexagonal structures formed by an array of cone-like pillars (Supplementary Fig. 1a, b and Fig. 1c). To mimic such a structure[22–24], we employ a unique 3D template method with high accurate parameter modification ability and potential for largescale production (Supplementary Fig. 2). For other functional layers, we fabricate the IENG by the layered self-assembly. The specific preparation method is described in the preparation process of the IENG below.

After fabrication, we found that the surface of the middle layer exhibits a periodic concave-convex structure that can trap more light for heat generation (Fig. 1d). We introduced a partial-processing method for pore fabrication in such a layer. Mesopores with diameters of approximately 20 nm are found dispersed evenly in this structure to guarantee a sufficient water supply during the evaporation process. The top layer is a grid-like porous structure made up of micro/ nanoscale porous nanofibers (Fig. 1e) with a thickness of 190 μm and a porosity of approximately 84.4% (Supplementary Fig. 3a). MWNT and MXene are added in the top and middle layer of the IENG (Fig. 1g) to decrease the internal resistance and increase the power output.

Fundamentally, the electricity generation of the IENG is a interaction process involving the choice of hydrophilicity, the surface charge density and device durablity. To demonstrate its potential as an excellent electric generator, we tested that the surface of the top layer is hydrophilic (Supplementary Fig. 3b), and the zeta potential is to be as high as −27.04 eV (Fig. 1f), which is the fundamental property for power generation. We also analyzed the surface charge density of this device under dry and wet conditions. As the layer changes from dry to wet, it sharply increases from a tiny negative charge (−0.43 nC cm$^{-2}$) to −14.2 nC cm$^{-2}$ (Supplementary Fig. 3c, d). In addition, we investigated the working stability of CsPbBr$_3$ perovskite in the middle layer of the IENG by leaving the device evaporating for 7 days. As the XRD spectrum shows, a clear characteristic peak after the test demonstrates excellent stability (Fig. 1h).

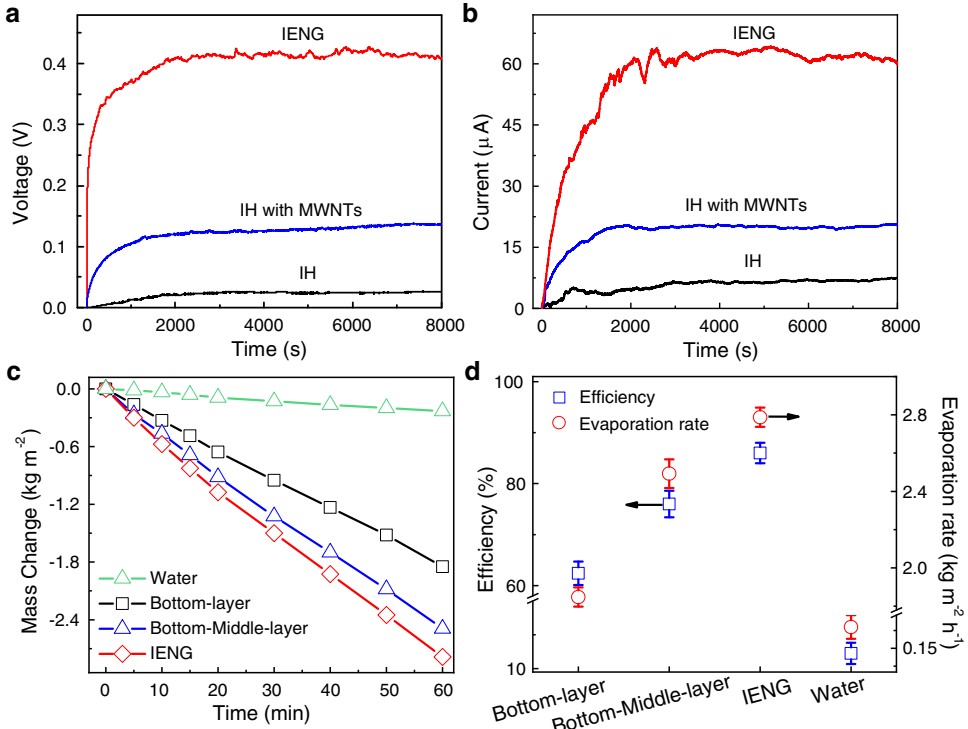

**Fig. 2 | Power generation performance and evaporation characteristics of the IENG. a** Open-circuit voltage per unit area of the IH, the IH with MWNTs and the IENG. **b** Short-circuit currents per unit area for the IH, the IH with MWNTs and the IENG. **c** Water evaporation rates of bulk water and the IENG with different layers in over 1 h. **d** Energy conversion efficiencies and water evaporation rates of bulk water and the IENG with different layers. The lengths, widths and heights were 10 mm, 10 mm and 20 mm, respectively. The error bars represent standard deviations.

## Power generation performance of the IENG induced by interfacial evaporation

Next, we tested the power generation property of our IENG under a light intensity of 2 kW·m$^{-2}$. As shown in Fig. 2a, b, an open-circuit voltage as high as 432 mV cm$^{-2}$ is reached, which is 9.82 times higher than the reported average value (Supplementary Fig. 4), and a short-circuit current is 64.2 μA cm$^{-2}$. We can also conclude that the superior performance originates from the MWNTs and the BL structure. When MWNTs are added to the IH, the open-circuit voltage and the short-circuit current greatly improved. Furthermore, when the BL structure is introduced, the output voltage and current sharply increase nearly 3 times. Consistent with the power output, the presence of the middle and top layers greatly improve the water evaporation performance of the device, the corresponding evaporation rate increases from 1.848 to 2.41 and 2.78 kg m$^{-2}$ h$^{-1}$, respectively. The calculated energy conversion efficiency of our IENG is 86.3% and the overall performance is higher than some existing solar evaporators (Fig. 2c, d, Supplementary Fig. 5a, b and Note 1, 2).

## Photothermal conversion effect on interface evaporation with bionic light-trapping structure

The photothermal conversion efficiency is the fundamental parameter for interfacial evaporators, which effectively affects the power generation performance. Our IENG is in accordance with bionic light-trapping structure. According to the equations given below,

$$f(x,y) = \begin{cases} 0 & x^2 + y^2 > d \\ n_2 - n_1 & x^2 + y^2 \le d \end{cases} \tag{1}$$

$$F(x,y) = n_1 + f(x,y) * \left[ \text{comb}\left(\frac{x}{\Lambda}\right) \text{comb}\left(\frac{y}{\Lambda}\right) \right] \tag{2}$$

where $n_2$ is the substrate's refractive index, $n_1$ is the refractive index of air, $d$ is the diameter of the cylinder, $f(x,y)$ is a single periodic function, and $*$ is the convolution symbol. In our IENG, the refractive index of the moth-eye structure is $n_2 = 30$ in the longwave infrared band. The microstructure period $\Lambda$ is $50 \pm 0.1\,\mu m$, the depth $h$ is $30 \pm 0.5\,\mu m$, and the bottom diameter $d$ is $20 \pm 0.1\,\mu m$. The theoretical reflectivity was 3% for our device (for the theoretical model, see Supplementary Fig. 6a)[25].

We measured the light absorbance of different samples at wavelengths between 190 and 2500 nm (Fig. 3a). The average light absorption efficiency of the bottom-middle layer is approximately 94.7%. The addition of CsPbBr$_3$ type perovskite (Supplementary Fig. 6b, c) and the porous nanofibers can further enhance the light absorption to 96.8%, which is approaching the theoretical reflectance values of moth eyes (~97%) and higher than those of other types of solar evaporators (Supplementary Fig. 6d). Such an excellent light absorption efficiency originates from multiple reflecting and absorption by the light absorption materials within the BL layer (Supplementary Fig. 7a). Consistent with the light spectral absorption results, our IENG's surface temperature was significantly improved compared with other samples (Fig. 3b, d). We also analyzed the side surface temperature of different samples by IR camera. The temperature shows a gradient distribution from the surface to the bottom of our IENG after irradiation for 1 h (Fig. 3c). It proves the outstanding ability of our IENG to locate the heat at the surface. The time-dependent side surface temperature variation at different depths for different samples was recorded in Fig. 3e. It showed that the side surface temperature of our IENG is significantly higher than that of the bottom-middle-layer at the same depth, further confirming our IENG's perfect light-to-heat conversion ability.

The water supply ability is also a critical bottleneck for evaporation. In the bottom middle layer of our IENG, the porous cellulose/ ionic liquid components can work as an enormous

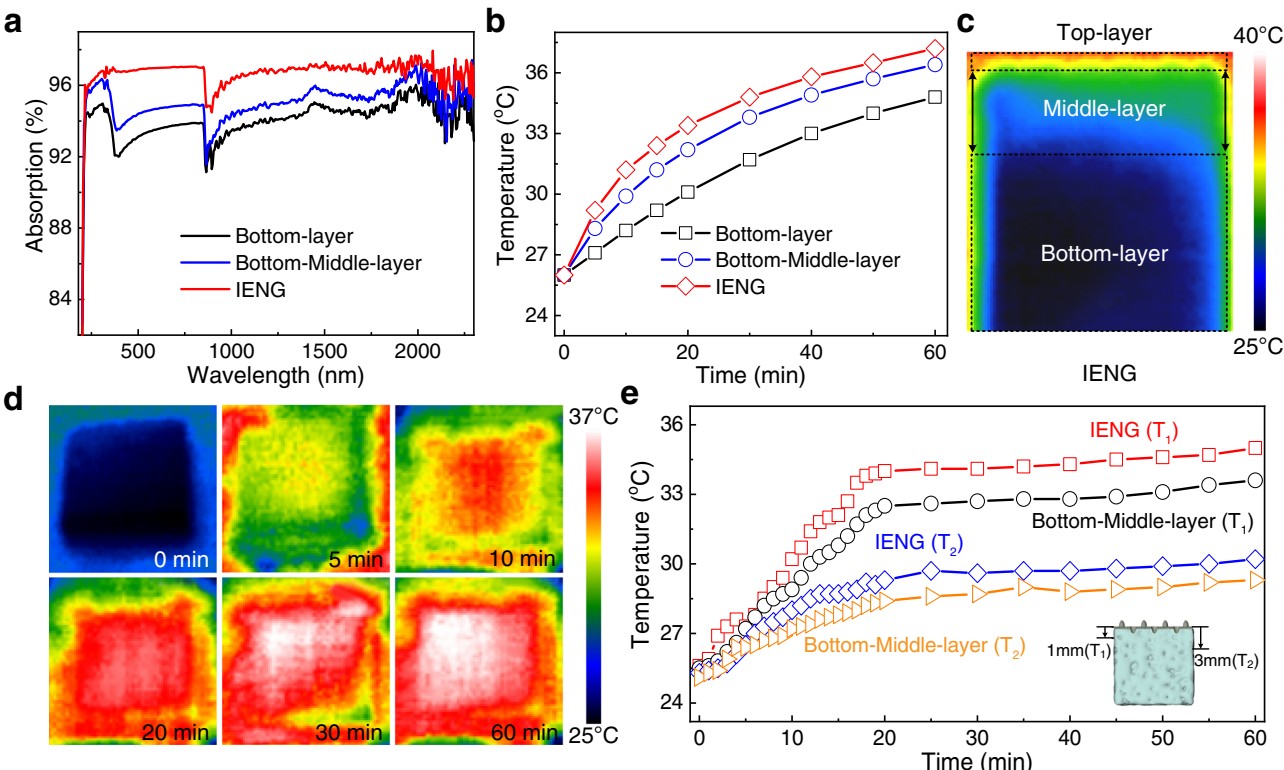

**Fig. 3 | Photothermal conversion characteristics of the IENGs. a** Light absorption spectra of the IENGs in the wavelength range of 190–2500 nm. **b** Surface temperature curves of the IENGs under a solar light intensity of 1.0 kW· m⁻². **c** Side surface temperature images of the IENG recorded via IR camera. **d** IR thermal imaging of the IENG evaporation interface. **e** Time-dependent side surface temperature variation at different depths for the IENG and the bottom-middle-layer.

reservoir and powerful pump[26,27]. We calculated different layers' saturated water content ratio to evaluate the water absorption capacity[28–30]. These results showed that the highest water content ratio was approximately 82% for the bottom-middle-layer (Supplementary Fig. 7b), guaranteeing water supply during evaporation.

In addition, we tested the vaporization enthalpy of our device, which showed a sharp decrease compared with pure water (Supplementary Fig. 8). That is because the hydrogel framework in our IENG disturbs the hydrogen bond net structure between water molecules, giving birth to plenty of intermediate water (IW) that consuming less energy when evaporation, thereby increasing the overall water evaporation rate[31,32].

### Power generation performance of the IENG in a marine environment

The marine environment varies all the time. Therefore, we analyzed the power generation of our device under different conditions, including the irradiation strength, ion concentration, and wind speed. First of all, we explored the power generation performance of the IENG with changing light intensity. By increasing the external light intensity from 0 to 2 kW·m⁻², the open-circuit voltage of the IENG increases from 0.09 to 0.42 V (Fig. 4a), and the short-circuit current increases from 12.5 to 122.3 μA (Fig. 4b), the maximum power density is tested to 5.1 μW·cm⁻² (Fig. 4c). At the same time, the water evaporation rate of the IENG is also improved with the improvement of light intensity (Supplementary Fig. 9a–c). Besides, we also found that the open-circuit voltage, short-circuit current and power density exhibit a simple exponential relationship with the light intensities (Supplementary Fig. 9d–i). To figure out the relation between water evaporation rate and the power output, we performed a linear fitting between them. As shown in Supplementary Fig. 9j–l, the power generation of the IENG increased

exponentially with water evaporation rate, which is in line with the theoretical formulas in Supplementary Note 3 and Note 4[32,33].

Then, we explored the power generation performance of the IENG in an environment of DI water, tap water, seawater and different concentrations of NaCl solution with a light intensity of 2 kW·m⁻². We found that the open-circuit voltage changing trend possesses two processes. It first increases from 0.42 for DI water to 0.493 V for sea water and then decreases to 0.434 V when further increasing the ion concentration to 1 mol L⁻¹ (Fig. 4d). That is because, for the first process, increasing ion concentration means more cations flowing simultaneously, thereby increasing open-circuit voltage. However, an excessive ion concentration would cause rapid salt precipitation during the evaporation process, blocking the internal pores for evaporation and decreasing the voltage output. At the same time, the increasing ions rapidly shield the surface charge of the IENG nanochannels, decaying the cation permselective ability of the IENG, also resulting in a deterioration in voltage output[34,35]. However, the short-circuit current continuously decreases from 122.3 μA to 113.1 μA and 103.2 μA (Fig. 4e). That is because the process of salt precipitation causes the internal resistance increasing of the IENG, so the short-circuit current of the IENG decreases continuously with the increase of the ion concentration. It should be noted, the influence of osmotic pressure caused by high ion concentration on its power generation is ignored here (Supplementary Fig. 10a, b). Under load resistance, the maximum power density of our IENG reached 5.9 μW·cm⁻² (Fig. 4f) for testing in seawater, making it possible for practical application (Supplementary Figs. 11 and 12).

Since wind speed strongly influences evaporation, we also explored the power generation performance of the IENG under 0, 1, 2 and 3 m s⁻¹ wind speed environments tested in seawater with a light intensity of 2 kW·m⁻². Figure 4g, h showed that as the wind speed increased from 0 to 1 m·s⁻¹, the open-circuit voltage of the

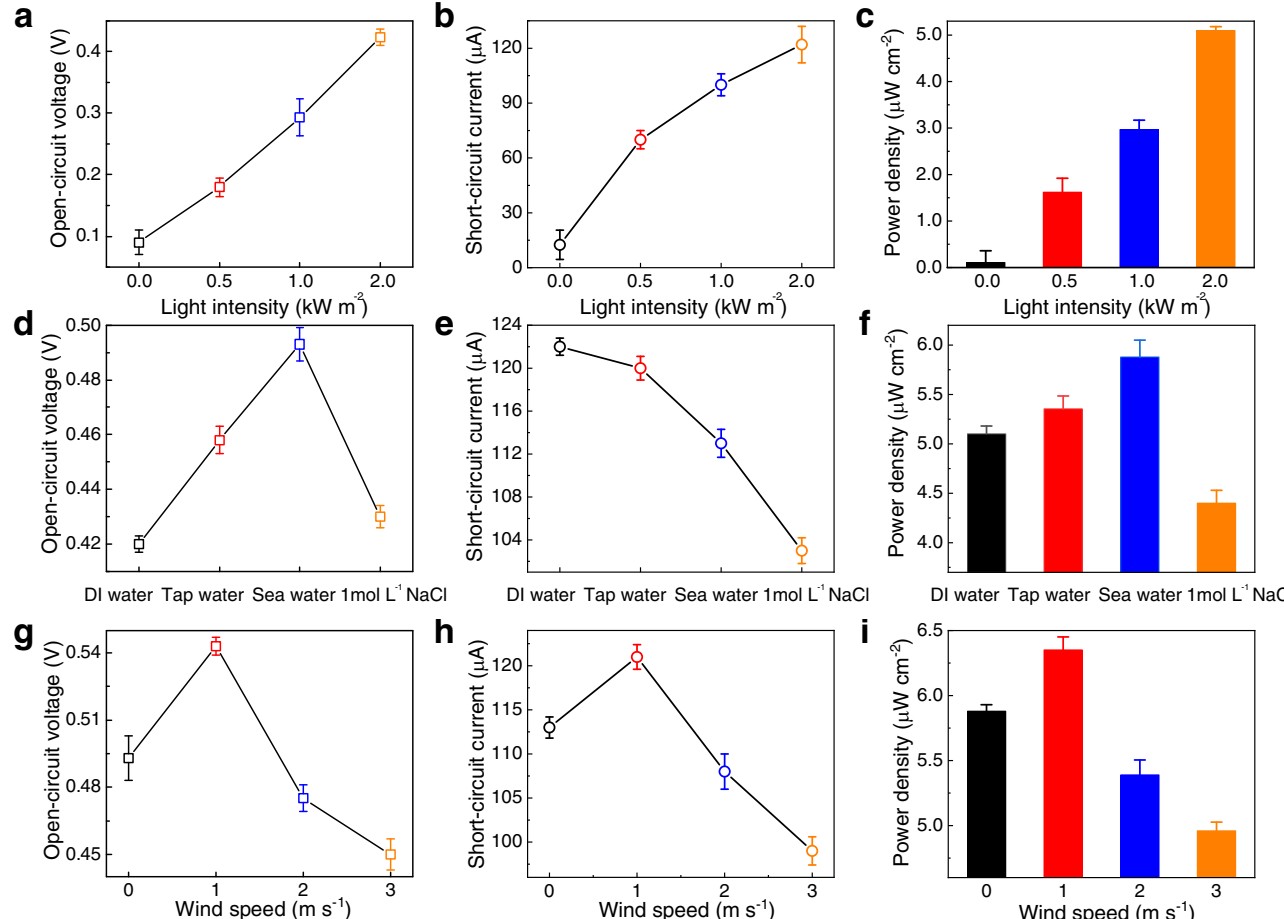

**Fig. 4 | Power generation performance of the IENG under different surrounding conditions. a–c** Power generation performance of the IENGs under different light intensities. **d–f** Power generation performance of the IENGs in different liquids.

**g–i** Power generation performance of the IENGs under different wind speeds. The lengths, widths and heights were 20 mm, 20 mm and 20 mm, respectively. The error bars represent standard deviations.

IENG increased from 0.493 to 0.543 V, and the short-circuit current increased from 113.1 to 121.4 μA. Under load resistance, its maximum power density reached 6.3 μW·cm$^{-2}$ (Fig. 4i). However, when the wind speed further increases beyond 1 m·s$^{-1}$, the power generation performance of the IENG begins to decrease. That is because when the wind speed is lower than 1 m·s$^{-1}$, the increasing wind speed benefits the water evaporation, improving the overall power generation performance of the IENG[4,36]. However, when the wind speed is too high, side wall evaporation dominates, weakening the interaction between the cation and the MWNTs buried mainly on the top surface[37].

These results illustrated that our IENG has optimized power generation performance in seawater with a light intensity of 2 kW·m$^{-2}$ and a wind speed of 1 m·s$^{-1}$. It had a maximum open-circuit voltage of 0.543 V, a maximum short-circuit current of 121.4 μA and an output power of 25.4 μW·with an external load of 7460 Ω.

## Working principle induced by interfacial evaporation and application of the IENG

The working principle of the IENG can be traced from two processes: (a) converting solar energy to kinetic energy of water molecules and (b) converting kinetic energy of water molecules to electricity.

In the former process, the nature-inspired moth's eye surface design significantly strengthened the light-to-heat conversion efficiency, increasing the water evaporation rate at the middle top interface under the sunlight. A huge transpiration pull force can be

generated as the water evaporation ($Q_{eva}$). At the same time, the ion concentration will also increase at the solid-liquid surface, contributing to an ever-growing osmosis force ($Q_{osm}$). These two forces in the same direction cooperate to quickly drag the water through the pores in the hydrogel and the gaps between the MWNTs[38,39] (Fig. 5a (i)).

The latter process can be explained by hydrovoltaic effect (Fig. 5a (ii)). Specifically, when contacting water, the oxygen-containing functional groups on the surface of the MWNTs, such as carboxyl and hydroxyl groups were hydrolyzed. Therefore, a negative charged electric layer is formed[40]. Then, cations ($H_3O^+$, $Na^+$, etc.) in water are attracted by this negative electric layer on the surface of the MWNTs through Coulomb force, and thus an electric double layer is formed. Due to the extremely narrow gap between the MWNTs, the Debye layers in the electric double layer are overlapped, where only cations dominate[41,42]. Therefore, when evaporation happens on the surface, water flows within the gaps between the MWNTs, dragging the $H_3O^+$ to the direction of water flow[43,44]. This caused a high concentration difference between the two ends of water flows, forming a flowing potential and a fluctuating Coulomb field[45]. After connection, the coupling of phonon wind and a fluctuating Coulomb field drove electron transfer to generate a direct current[46].

The water evaporation rate of our IENG reaches 4.385 kg m$^{-2}$ h$^{-1}$ with a 2.0 kW·m$^{-2}$ light irradiation and a 1 m s$^{-1}$ wind speed (Fig. 5b). The freshwater quality met the WHO's human drinking water standards (Supplementary Fig. 13a)[47,48]. Besides, our IENG can perform well in the actual environment, proving its practicability (Supplementary

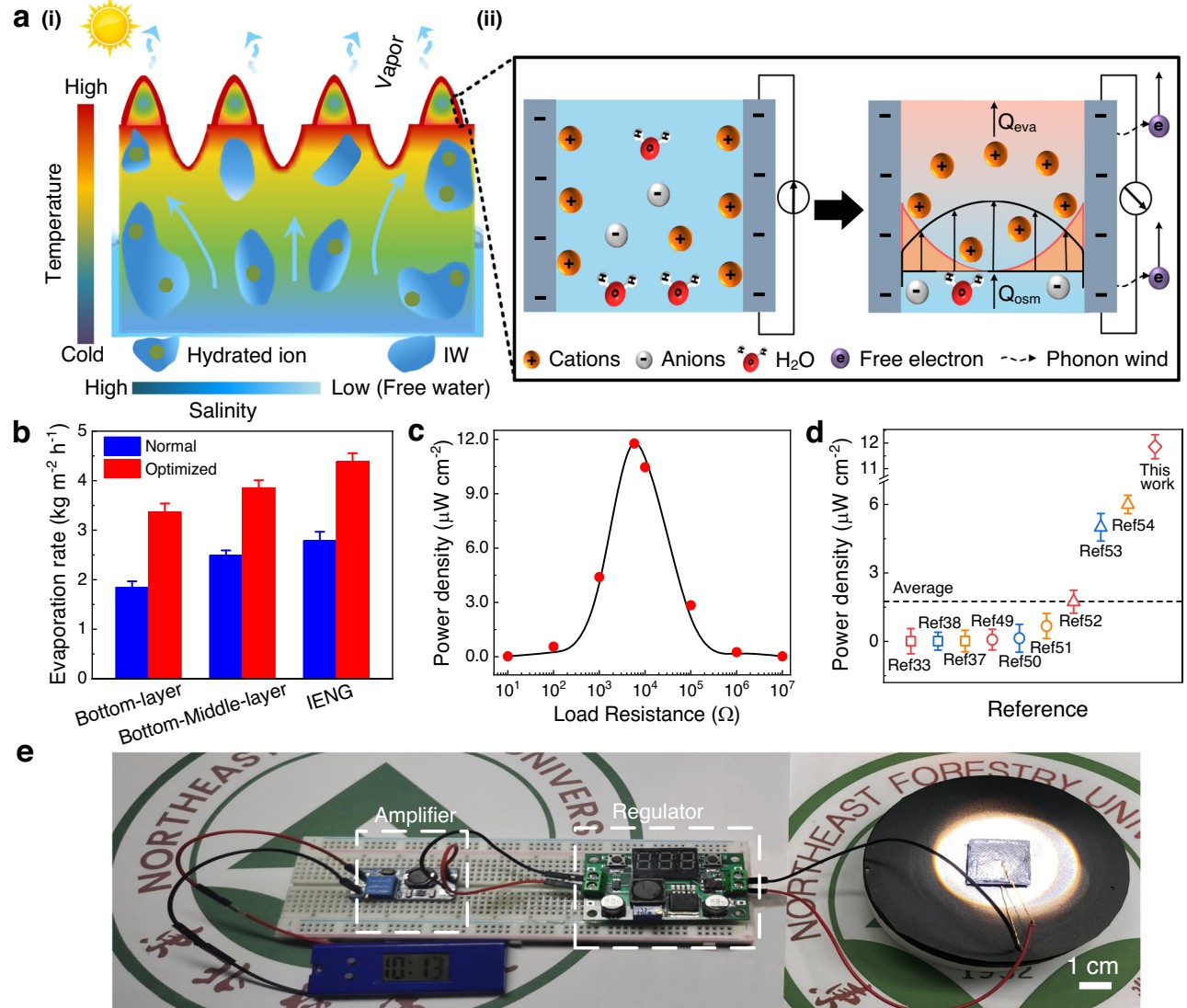

**Fig. 5 | Working principle and application of the IENG. a** Power generation principle for the IENG induced by water evaporation: (i) Schematic diagram of water flowing path and evaporation in the IENG, (ii) The left represents schematic diagram of the initial stage, which depicts the specific ion distribution within the channel. And the right is the schematic diagram of the steady state, which depicts the ion transportation in the overlapping electric double layer. **b** Water evaporation rate of the IENG in a simulated marine environment. The normal environment had a

1.0 kW· m⁻² light intensity, no wind and deionized water, and the optimized environment had a 2.0 kW· m⁻² light intensity, a 1 m s⁻¹ wind speed and seawater. **c** Output power density of the IENG measured under different load resistances. **d** Comparison of power density generated by the IENG and other solar-driven IENGs. **e** Photograph of a developed self-powered working system. The error bars represent standard deviations.

Fig. 14). When the load resistance reaches 5793 Ω, the loaded output power reaches a maximum of 11.8 µW cm⁻² (Fig. 5c and Supplementary Fig. 13b), more than 6.8 times higher than the currently reported average value[33,37,38,49–54] (Fig. 5d).

We also did a statistical analysis. With a freshwater production of 2.19 kg h⁻¹ m⁻³ and an electrical generation of 10 W h m⁻³, our IENG device only costs approximately 800 RMB m⁻² showing its potential in realisticity and scalability. Furthermore, we designed a self-powered electronic integrated system. As shown in Fig. 5e, a low-voltage device can operate continuously with only solar and wind energy input, demonstrating an opportunity to develop offshore power generation platforms and freshwater supply devices (working system circuit in Supplementary Fig. 15a, b).

## Discussion
In summary, we have demonstrated an efficient multi-layered interfacial evaporation-driven power generation system

mimicking the natural structure of the moth's eye surface. The advantages of this device include excellent moisture storage/supply ability, outstanding solar-heat-conversion property and remarkable electric conductivity, which allows it to efficiently utilize the ambient low-grade heat for freshwater collection and power generation. Under the modified condition, our device performs an excellent freshwater production of 2.78 kg m⁻² h⁻¹ with a light intensity of 1.0 kW·m⁻² and an electric output power density of 11.8 µW cm⁻², which is over 6.8 times larger than the average devices. The synergistic effect from enhanced evaporation and hydrovoltaic effect contributes to such a good performance. Therefore, this work demonstrates a sustainable interfacial evaporation-driven power generation approach and provides a foundation for utilizing multiple natural energy sources. It can serve to develop offshore power generation platforms and freshwater supply devices in the future.

## Methods

### Materials

The ionic liquid of 1-n-butyl-3-methylimidazolium chloride (BMIMCl) used in the experiment was purchased from Lanzhou Institute of Chemical Physics, Chinese Academy of Sciences. Polyethylene glycol (PEG, molecular weight 100 MW) and $CsPbBr_3$ type perovskite were purchased from Macleans Co., Ltd, China. α-Cellulose (90 μm particle size), ethyl cellulose and chitosan (Cs) were obtained from Aladdin Co., Ltd, China. Acetic acid (purity 60%), ethanol and polyaniline were purchased from Yongchang Reagent Co., Ltd, China. MWNTs (diameter 15 nm, ion concentration 10 wt %) and MXene (molecular weight 194.6) were purchased from Turing Technology Co., Ltd, China.

### Preparation process of the IENG

First, cellulose were dissolved in BMIMCl under water bath conditions. MWNTs was added to the mixture in sequence and stirred evenly. The 2/3 volume of the mixture was poured directly into a special mold of polylactic acid, and the 1/3 volume of the mixture was poured into the top of this mold by adding $MXene/CsPbBr_3$ and mixing evenly. After phase separation, the mixture was placed and cooled, whose upper surface has the designed bionic light-trapping structure. A mesoporous IH with bionic light-trapping surface (the bottom-middle-layer) is prepared. Afterwards, acetic acid solution dissolving with Cs, PEG and MWNTs dispersions to prepare precursor solution is obtained for electrospinning. The precursor solution was stirred and injected into a JDF05 electrostatic spinning machine with 20 kV voltage and 0.1 mm s$^{-1}$ pressure velocity. It was sprayed on the surface of bionic light-trapping layer by a self-assembly method to construct the top layer. After drying, the IENG with the natural surface structure of moth's eye was fabricated.

### Materials characterization

The structural characteristics of the IENG were observed through a scanning electron microscope with an accelerating voltage of 5 kV (JSN-7500F, Japan). The optical transmittance and the reflectance spectrum of the IENG were measured with an ultraviolet-visible spectrophotometer (Lambda 950, USA) in the range of 190-3000 nm. The light absorption efficiency was calculated by the Equation A = 1-R-T, where R and T are the reflection and transmission efficiencies of the IENG, respectively. Infrared photos of the IENG were taken by an HT-18 infrared camera. The evaporation experiment was carried out under a solar simulator of a CEL-SA500/350 xenon lamp. Ion concentration was tested using an inductively coupled plasma spectrometer (ICP-OES) and an ion chromatograph instrument (ICS-600). The power generation of the IENG was measured by a Keithley 6514 electrometer (USA).

### Measurement of solar vapor generation

The IENG was placed in an experimental cistern (30 °C, 40% RH, summer, in Harbin, China). A solar vapor evaporation device was placed on an electronic balance and illuminated by solar simulator to monitor evaporation quantity in real time. The mass change of sea water in the solar evaporator was recorded transiently by an electronic balance.

### Measurement of power generation

The power generation measurement used the solar vapor evaporation device to supplement wind energy and other modules to simulate marine environment (21.4 °C, 15.8% RH, winter, in Harbin, China). Before the electrical performance test, a stable conductive system was constructed by intermittently dropping polyaniline/ ethanol on the surface. The short-circuit current test is concentrated on the depth of the boundary of the middle-top-layer interface, and the open-circuit voltage test is connected to the top layer surface and approximately the middle height of the middle layer. Here, gold electrodes were selected to test the electricity performance (Supplementary Fig. 16). The upper and lower test positions need to be dynamically adjusted at the first time. The power generation performance of the IENG was measured after connected to a Keithley 6514 electrometer (USA) under different light intensities by solar simulator.

## Data availability

The data that support the plots within this paper and other findings of this study are presented in the main article and the supplementary materials. Additional data related to this paper may be requested from the corresponding authors upon reasonable request.

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

## Acknowledgements

This work is supported by National Natural Science Foundation of China (Grant No. 51905085, 31925028, 52175266, 51975502), China Post-doctoral Science Foundation Funded Project (Grant No. 2018M630330 & No. 2019T120245), and Shenzhen Science and Technology Innovation Council (SGDX2020110309300502).

## Author contributions

Z.S. conceived the idea. Z.L. designed the experiment and analyzed the data, Z.S., H.Y., M. J. provide financial support. H.Y. and Z.W. provided technical support for cellulose and experiments. Z.S., Z.L., C.H., S.G. wrote and revised the article, and Z.W., Z.S. H.Y. conducted the final review and revision version of this article. All authors discussed the results and commented on the manuscript.

## Competing interests

The authors declare no competing interests.
