## [Peer Review File · Nature Communications]

Achieving efficient power generation by designing bioinspired and multi-layered interfacial evaporatorREVIEWER COMMENTS

Reviewer #1 (Remarks to the Author):

In this paper, the authors demonstrated an interfacial solar vapor evaporation material which could simultaneously generate electricity. Though this concept is of some interests, the working mechanisms of this material are not well understood in this manuscript. The authors added so many compounds in their hybrid material and told a long story about the functions of each component. However, I see few experimental results or references to support their claims. Besides, the relationship between solar vapor evaporation and power generation was not clear. In addition, this manuscript was poorly written with many grammar errors and odd statements. Thus I don't recommend its publication in Nature Communications. More detailed concerns are described below:

1. The Introduction part was poorly written. The authors didn't mention any previous studies on solar vapor generation or related power generation. What's the state-of-art in this area? I also found an odd statement, "The all-inorganic-type perovskite (Cs_4PbBr_6) with a crystal structure similar to the moth eye structure". The crystal structure describes the arrangement of atoms. How can it be similar to the moth eye? Did you use moth eye structured perovskite in your study?
2. The authors claimed that perovskite particles could help absorb photons and improve the temperature of interfacial layer. Do you have any references to support this claim? Or could you compare the temperature of interfacial layer with or without perovskite particles under light to prove this? In addition, perovskite is known to be sensitive to moisture and unstable in ambient environment. Can you comment on the stability of your particles in this evaporator?
3. The authors claimed that the TiO_2 and SiO_2 particles absorbed light energy to generate electron-hole pairs. In this case, this part of energy is wasted. Even the energy is released after recombination of electrons and holes, this part energy is dissipated in the hydrogel matrix and not useful to heat the interfacial evaporating layer.
4. What's the exact role of ionic liquid in this hybrid material? How can the concentration of BMIM⁺ ions affect the solar vapor and power generation properties? Could you use other ions to replace the BMIM⁺ ions?
5. According to my understanding, the power generation in this evaporator is caused by the porous MWNT layer, which acts as an osmotic power generator. Thus the properties of this MWNT layer is critical important in this material. However, the authors didn't provide any characterizations on this part. What's the thickness? What's the porosity? What are the surface properties? How is the charge distribution on surface?
6. The osmotic power generation is affected by the ion concentrations. The authors also found that the power generation properties were changed in saline water. So how does the ion concentration of saline water affect the power generation in this material? Will Na⁺ or Cl⁻ ions accumulate in the MWNT layer?

7. The authors claimed that the enthalpy of water evaporation was changed in this evaporator. What's the enthalpy change? Without this information, how can you calculate the energy conversion efficiency?
8. What's the relationship between vapor generation rate and power generation? Could you measure these values under different light intensities?
9. The authors measured the properties of their material in marine environment under 2 kW/m² light intensity. How can you get such high light intensity in marine environment? The solar intensity on earth surface is usually under 1 kW/m².
10. Panel g and f in Supplementary Figure 2 are missing.
11. So many details are missing in the Methods Section. The authors should carefully describe how they conduct the solar vapor generation and power generation measurements.

Reviewer #2 (Remarks to the Author):

Solar powered interfacial evaporators have a great potential in delivering clean water and electric power simultaneously from a single device. Such hybrid devices would be valuable everywhere from crowded cities to remote areas where the use of land is limited, or electricity and clean water are not readily available. New technologies are needed to harness both forms of energy as efficiently as possible, ideally at low cost. Sun and co-workers have developed a 3D structured interfacial evaporator mimicking the natural surface structure of moth's eye. The device not only delivered a respectable evaporation rate, but also had a secondary function of driving a nanogenerator, meaning that the evaporation process itself generated direct current. In my view, the biggest value of this work comes from the nature-inspired moth's eye design that improved light trapping and thereby the overall efficiency of the hybrid device. This intriguing system is presented in an interesting way considering the readership of the journal, making a significant impact in the field. This is a solid piece of work with comprehensive characterization and data analysis. However, this work has potential for even better in terms of readability and rigorous grammatic revision to meet the standards of Nature Communications. Some general comments to the authors:

- 1) The title in the SI file is different from the main manuscript. Please correct.
- 2) It is not clear to me what the authors want to say with the phrase "greatly potential for sustainable spontaneously power generation" in the abstract and later in the text. Could you please clarify? Similarly, some sentences in the abstract are rather long and hard to follow, such as "Simultaneously, inspired by the light-trapping properties of moth eye, a simple and efficient BLT-

IENG including a hierarchical surface of bionic light-trapping and electrospinning perovskite conductivity with an enhanced thermally insulating and water storage capability is designed.” Also, “This work provides an unexplored strategy for multi-energies inspired natural interfacial evaporation driven power generation.” I would recommend splitting the sentences shorter and revising them so that they read better.

3) At the end of page 2, the authors write “...large thermal insulating capacity (increased by 1.5 times)”. Please specify what is that 1.5 times increase compared to. On pages 3–4, please describe what are “control 1” and “control 2”, where first mentioned.

4) The text in Fig. 5b,c is hard to read. Please increase the text size.

5) Related to one of the comments above, I would suggest shortening and clarifying some of the sentences in the conclusion as well. Such as “In summary, we demonstrated a solar and wind natural energies driven interfacial evaporation nanogenerator (BLT-IENG) including a hierarchical surface of bionic light-trapping and electrospinning perovskite conductivity with an enhanced thermally insulating and water storage capability.”

6 It would very be attractive to see discussion about the scalability and cost-effectiveness of the moth’s eye design and the chosen perovskite, carbon nanotube, etc. materials. Surely, the cost of the devices plays another important role when thinking about their potential use. Is there anything to develop further in that regard? At least as “future perspectives” in the conclusion section. How would the water evaporation rate and power output scale in realistic conditions?

Response Letter for

**Achieving efficient power generation by designing bioinspired
and multi-layered interfacial evaporator**

Zhuangzhi Sun^{1,2,}, Chuanlong Han¹, Shouwei Gao³, Zhaoxin Li^{1,*}, Mingxing Jing¹, Haipeng Yu^{2,*},
Zuankai Wang^{3,*}*

¹ Province Key Laboratory of Forestry Intelligent Equipment Engineering, College of Mechanical and Electrical Engineering, Northeast Forestry University, Harbin 150000, People's Republic of China.

² Key Laboratory of Bio-based Material Science & Technology, Ministry of Education, Northeast Forestry University, Harbin 150000, People's Republic of China.

³ Department of Mechanical and Biomedical Engineering, City University of Hong Kong, Hong Kong, People's Republic of China.

*E-mail: sunzhuangzhi@nefu.edu.cn; 2014211213@nefu.edu.cn; yuhaipeng20000@aliyun.com;
zuanwang@cityu.edu.hk*

Itemized list of response to the reviewers' remarks
(Black: Reviewers' remarks; Blue type: Our response)

Reviewer #1 (Remarks to the Author):

In this paper, the authors demonstrated an interfacial solar vapor evaporation material which could simultaneously generate electricity. Though this concept is of some interests, the working mechanisms of this material are not well understood in this manuscript. The authors added so many compounds in their hybrid material and told a long story about the functions of each component. However, I see few experimental results or references to support their claims. Besides, the relationship between solar vapor evaporation and power generation was not clear. In addition, this manuscript was poorly written with many grammar errors and odd statements. Thus I don't recommend its publication in Nature Communications. More detailed concerns are described below:

Response: First of all, we are very grateful to the reviewers for the valuable and constructive comments, which is of great significance for improving the quality of our article. We have tried our best to address the raised questions one by one in the response below, and revise our manuscript accordingly. In response to the above problems, we re-drew the schematic diagram of water evaporation-driven power generation (Fig.5a) and described the working principle of the IENG (*Response to (1)*). As suggested by reviewers, we simplified the materials composition of this device, highlighted the main innovation/ claims, and complemented supported experiments. All the claims and experimental evidence mentioned are listed in *Response to (2)*. We have supplemented the experiments to reveal relationship between water vapor evaporation and power generation under different light intensities (*Response to (3)*). English language polishes were performed by Springer Nature Authors Services (Language Paper ID: FE37-01B7-7B75-0606-7ED4 & E5F6-59BC-3609-D23E-96CC & F778-F8A9-0A1F-8182-A94B) to make the statement more fluent and smooth (*Response to (4)*). Simultaneously, we have implemented major revisions to rewrite this article and made our description of the innovations clearer and accurate.

Additionally, in response to the reviewers' comments, we conducted an extensive literature search. In this process, we found that the experimental error may be caused by electrochemical corrosion or polarization of test electrodes (*Nat. Nano.* 12(4): 317-321 (2017). *Angew. Chem. Int. Edit.* 132(26): 10706- 10712. (2020). *ACS Appl. Mater.*

Inter. 12(9): 11232-11239 (2020)). In order to make our experimental results more rigorous and comparable, we replaced the original copper (Cu) wrapped with tin (Sn) electrodes (~ 1.1 V, ~ 220 μA , 35.14 $\mu\text{W cm}^{-2}$) with gold (Au) electrodes (543 mV, ~ 121.4 μA , 11.8 $\mu\text{W cm}^{-2}$), and re-executed the power generation experiment with Au test electrodes in Fig. 2, Fig. 4 and supplementary materials. Here, it should be explained that the difference in power generation performance in IENG produced by test electrode materials will be further studied in our follow-up work.

We believe that these above results are more objectively to reflect the performance and advantages of our IENG. Therefore, we revised the full manuscript according to the reviewer's suggestion and combined with the above ideas. Thank you so much for your understanding. We sincerely hope an opportunity for us to reconsider our work, and we also sincerely invited you to review our revised work again. The detailed modification content is shown below.

(1) Though this concept is of some interests, the working mechanisms of this material are not well understood in this manuscript.

Response to (1): Thanks so much for your constructive suggestion. The revised working principle of our IENG and the detailed description of evaporation-driven power generation of the IENG are as follow.

The working principle of the IENG can be traced from two processes: (a) converting solar energy to kinetic energy of water molecules and (b) converting kinetic energy of water molecules to electricity.

In the former process, the nature-inspired moth's eye surface design significantly strengthened the light-to-heat conversion efficiency, increasing the water evaporation rate at the middle-top-interface under the sunlight. A huge transpiration pull force can be generated as the water evaporation (Q_{eva}). At the same time, the ion concentration will also increase at the solid-liquid surface, contributing to an ever-growing osmosis force (Q_{osm}). These two forces in the same direction cooperate to quickly drag the water through the pores in the hydrogel and the gaps between the MWNTs^{1,2} (Fig. 5a (i)).

The latter process can be explained by hydrovoltaic effect (Fig. 5a (ii)). Specifically, when contacting water, the oxygen-containing functional groups on the surface of the MWNTs, such as carboxyl and hydroxyl groups were hydrolyzed. Therefore, a negative charged electric layer is formed³. Then, cations (H_3O^+ , Na^+ , etc.) in water are attracted by this negative electric layer on the surface of the MWNTs through Coulomb force,

and thus an electric double layer is formed. Due to the extremely narrow gap between the MWNTs, the Debye layers in the electric double layer are overlapped, where only cations dominate^{4,5}. Therefore, when evaporation happens on the surface, water flows within the gaps between the MWNTs, dragging the H_3O^+ to the direction of water flow^{6,7}. This caused a high concentration difference between the two ends of water flows, forming a flowing potential and a fluctuating Coulomb field⁸. After connection, the coupling of phonon wind and a fluctuating Coulomb field drove electron transfer to generate a direct current⁹.

Fig. 5 Working principle and application of the IENG. a Power generation principle for the IENG induced by water evaporation: (i) Schematic diagram of water flowing path and evaporation in the IENG, (ii) The left represents schematic diagram of the initial stage, which depicts the specific ion distribution within the channel. And the right is the schematic diagram of the steady state, which depicts the ion transportation in the overlapping electric double layer.

(2) The authors added so many compounds in their hybrid material and told a long story about the functions of each component. However, I see few experimental results or references to support their claims.

Response to (2): Thanks so much for your constructive suggestion. Here, we reinterpreted the innovation in this work. We are extremely sorry that in order to avoid missing details affecting readability, we told a long story and made many claims that affected the logical structure of this article. We carried out a major change to the article and placed some supporting claims to supplementary materials. In addition, in order to facilitate to review, the main innovations, supporting claims and the corresponding experimental evidence of this article are listed below.

Main innovation:

- (a) An multi-layered interfacial evaporation-driven nanogenerator is developed by introducing bionic light-trapping structure for efficient light to heat and power generation.
- (b) This device performs a spectacular continuous power output as high as $11.8 \mu\text{W cm}^{-2}$ under optimal conditions, more than 6.8 times higher than the currently reported average value (Fig. 5d).

Fig. 5d Comparison of power density generated by the IENG and other solar-driven IENGs.

Supporting claim 1 for innovation (a): Our design of the middle and top layer harnesses the light-trapping structure on the surface of a moth's eye that exhibits nearly zero reflection of sunlight for evaporation.

Proof:

- (a) The effect and stability of the light absorption are further improved (Fig. 3a).
- (b) This results in an average absorption efficiency of approximately 96.7%, which is higher than that of other types of solar evaporators (Supplementary Fig. 6d).
- (c) Consistent with the light spectral absorption results, the surface temperature of the IENG (Fig. 3b-d) are significantly improved.
- (d) We investigated the working stability of CsPbBr₃ perovskite in the middle layer of the IENG by leaving the device evaporating for 7 days. As the XRD spectrum shows, a clear characteristic peak after the test demonstrates excellent stability (Fig. 1h).
- (e) Consistent with the power output, the presence of the middle and top layers greatly improve the water evaporation performance, the corresponding evaporation rate increases from 1.848 to 2.41 and 2.78 $\text{kg m}^{-2} \text{h}^{-1}$, respectively (Fig. 2c).
- (f) The calculated energy conversion efficiency of our IENG is 86.3% and the overall performance is higher than some existing solar evaporators (Fig. 2d, Supplementary Fig. 5a, b and Note1, 2).

Supporting claim 2 for innovation (a): The bottom layer comprises ionic hydrogel (IH), which is the stable water storage/ supply during evaporation.

Proof:

- (a) The enthalpy change in the water evaporation is reduced, thereby increasing the water evaporation rate (Supplementary Fig. 8).
- (b) The water absorption capacity of the bottom-middle-layer can be calculated for the saturated state. Results showed that the highest water content ratio is approximately 82% (Supplementary Fig. 7b).
- (c) The IENG showed an excellent water evaporation rate ($2.78 \text{ kg m}^{-2} \text{ h}^{-1}$) with a light intensity of $1.0 \text{ kW} \cdot \text{m}^{-2}$ (Fig. 2c).

Supporting claim for innovation (b): Resulting from the high evaporation rate and electric parameters, our IENG exhibits an excellent power density under the modified condition, more than 6.8 times higher than the currently reported average value.

Proof:

- (a) MWNT and MXene are added in the top and middle layer of the IENG to decrease the internal resistance of the device and increase the power output (Fig. 1g).
- (b) The surface of the top layer is hydrophilic (Supplementary Fig. 3b), and the zeta potential is to be as high as -27.04 eV (Fig. 1f), which is the fundamental property for electric generation.
- (c) We analyzed the surface charge density of this device under dry and wet conditions. As the layer changes from dry to wet, it sharply increased from a tiny negative charge (-0.43 nC cm^{-2}) to -14.2 nC cm^{-2} (Supplementary Fig. 3c, d).
- (d) When MWNTs were added to the IH, the open-circuit voltage and the short-circuit current greatly improved. Furthermore, when the BL structure was introduced, the output voltage and current sharply increase nearly 3 times (Fig. 2a,b).
- (e) Our IENG has optimized power generation performance in sea water with a light intensity of $2 \text{ kW} \cdot \text{m}^{-2}$ and a wind speed of $1 \text{ m} \cdot \text{s}^{-1}$. It had a maximum open-circuit voltage of 0.543 V , a maximum short-circuit current of $121.4 \text{ } \mu\text{A}$ and an output power of $25.4 \text{ } \mu\text{W}$ with an external load of $7460 \text{ } \Omega$ (Fig. 4, Supplementary Fig. 9-12).
- (f) When the load resistance reaches $5793 \text{ } \Omega$, the loaded output power reaches a maximum of $11.8 \text{ } \mu\text{W cm}^{-2}$ (Fig. 5c and Supplementary Fig. 13b), more than 6.8 times higher than the currently reported average value (Fig. 5d).

(3) Besides, the relationship between solar vapor evaporation and power generation was not clear.

Response to (3): Thanks so much for your constructive suggestion. We supplemented the experiments of water evaporation and power generation with the change of light intensity, and analyzed a relationship between water evaporation and power generation under different light intensities. The description of the relationship between water evaporation and power generation is as follows.

In this article, the open-circuit voltage, short-circuit current and output power density of the IENG under different light intensities were provided in Fig. 4a-c. As suggested by the reviewers, we supplemented experiments of water evaporation rate and power generation of the IENG with the change of light intensity, and revealed the relationship between water evaporation rate and power generation under different light intensities. The detailed experimental results are shown in Supplementary Figure 9. The specific analysis is as follows.

When the external conditions remain unchanged, it can be found that by continuously increasing the ambient light intensity of 2 kW m^{-2} , water evaporation rate of the IENG has been greatly increased to $4.385 \text{ kg m}^{-2} \text{ h}^{-1}$ (Supplementary Figure 9a, b). Through the analysis, it can be found that water evaporation rate of the IENG has an exponential function relationship with the light intensity (Supplementary Figure 9c). This is mainly because as the light intensity increases, the energy absorbed by the IENG increases, which in turn increases its water evaporation rate. However, since water evaporation rate of the IENG is also limited by water absorption properties of its materials, and it has a maximum value. Meanwhile, it is also found that the open-circuit voltage and the short-circuit current generated by the IENG increase with the increasing intensity of light received. By testing the open-circuit voltage and the short-circuit current generated by the IENG under different light intensities, it is found that the open-circuit voltage and the short-circuit current generated by the IENG increase with the increase of the light intensity (Supplementary Figure 9d, f). This is because the open-circuit voltage and the short-circuit current generated by the IENG are mainly produced by its water evaporation process. Therefore, the relationship between the open-circuit voltage and the short-circuit current of IENG and the change of light intensity is similar to the relationship between water evaporation rate and the light intensity (Supplementary Figure 9e, g). By testing the maximum output power of the IENG under different light

intensities, it can be found that the output power density also increases with the increase of light intensity (Supplementary Figure 9h). The output power density of the IENG also has an exponential relationship with light intensity (Supplementary Figure 9i).

In addition, combining with the open-circuit voltage, the short-circuit current and the power density analysis under different light intensities, a relationship of the IENG between power generation and water evaporation rate under different light intensities is obtained (Supplementary Fig. 9j-l). The power generation of the IENG increases exponentially with the improvement of its water evaporation rate under different light intensities, which is in line with the theoretical formulas obtained in Supplementary Note 3 and Supplementary Note 4 (*ACS Appl. Mater. Inter.* 2020, 12(9): 11232-11239).

Supplementary Figure 9. Relationship of the power generation performance, light intensity and water evaporation rate. **a** Mass loss of the IENG under different light intensities. **b** Evaporation rate of the IENG under different light intensities. **c** Relationship between the evaporation rate and the light intensity. **d** Open-circuit voltage under different light intensities **e**

Relationship between the open-circuit voltage and the light intensity. **f** Short-circuit-current under different light intensities. **g** Relationship between the short-circuit current and the light intensity **h** Power density under different light intensities. **i** Relationship between the power density and the light intensity. **j-l** Relationship between power generation and water evaporation rate under different light intensities.

Fig. 4 Power generation performance of the IENG under different surrounding conditions. **a-c** Power generation performance of the IENGs under different light intensities. **d-f** Power generation performance of the IENGs in different liquids. **g-i** Power generation performance of the IENGs under different wind speeds. The lengths, widths and heights were 20 mm, 20 mm and 20 mm, respectively.

(4) In addition, this manuscript was poorly written with many grammar errors and odd statements.

Response to (4): Thanks so much for your constructive suggestion. We have re-edited language and grammar in full article three times by Springer Nature Authors Services (Language Paper ID: FE37-01B7-7B75-0606-7ED4 & E5F6-59BC-3609-D23E-96CC & F778-F8A9-0A1F-8182-A94B) to increase its readability and meet journal requirements.

1. The Introduction part was poorly written. The authors didn't mention any previous studies on solar vapor generation or related power generation. What's the state-of-art in this area? I also found an odd statement, "The all-inorganic-type perovskite (Cs₄PbBr₆) with a crystal structure similar to the moth eye structure". The crystal structure describes the arrangement of atoms. How can it be similar to the moth eye? Did you use moth eye structured perovskite in your study?

Response: Thanks so much for your constructive suggestion. We have supplemented the stat-of-art on solar vapor generation, and the specific modification is described as follows, which is highlighted in the manuscript on page 2.

Solar-heat-driven interfacial evaporation has been identified as a promising green and sustainable solution for the pressing global problem of water shortage which can directly transfer the light to heat for evaporation¹⁰⁻¹³. With an elegant choice of materials, conditions, and structures, the evaporation rate can reach over 4 kg m⁻² h⁻¹ under 1 sun¹⁴⁻¹⁶. The light absorption efficiency of the surface is the fundamental bottleneck that restrains further increase in the evaporation performance.

For the description of perovskite, we extremely apologize for this inaccurate statement. Here, we have corrected this statement and deleted the wrong description in the article. It is necessary to clarify that the biological template method is used to construct bionic light-trapping structure, not perovskite in this work. This revision expression has been corrected in the manuscript.

Supplementary referee for the full article

10. Zhao, F. et al. Materials for solar-powered water evaporation. *Nat. Rev. Mater.* **5**, 388-401 (2020).
11. Sun, Z. et al. A high-efficiency solar desalination evaporator composite of corn stalk, Mcnts and TiO₂: ultra-fast capillary water moisture transportation and porous bio-tissue multi-layer filtration. *J. Mater. Chem. A* **8**, 349-357 (2020).
12. Zhu, L. et al. Recent progress in solar-driven interfacial water evaporation: advanced designs and applications. *Nano Energy* **57**, 507-518 (2019).
13. Zhang, Y. et al. Hierarchically structured black gold film with ultrahigh porosity for solar steam generation. *Adv. Mater.* **21**, 2200108 (2022).
14. Guo, Y. et al. Biomass-derived hybrid hydrogel evaporators for cost-effective solar water purification. *Adv. Mater.* **32**, 1907061 (2020).
15. Zhou, X. et al. Topology-controlled hydration of polymer network in hydrogels for solar-driven wastewater treatment. *Adv. Mater.* **32**, 2007012 (2020).
16. Guo, Y. H. et al. Hydrogels and Hydrogel-Derived Materials for Energy and Water Sustainability. *Chem. Rev.* **120**, 7642-7707 (2020).

2. The authors claimed that perovskite particles could help absorb photons and improve the temperature of interfacial layer. Do you have any references to support this claim? Or could you compare the temperature of interfacial layer with or without perovskite particles under light to prove this? In addition, perovskite is known to be sensitive to moisture and unstable in ambient environment. Can you comment on the stability of your particles in this evaporator?

Response: Thanks so much for your constructive suggestion. Massive literatures indicated that there is no doubt that perovskite has excellent light absorption capacity (*Angew. Chem.* 2015, 127(19): 5785-5788. *ChemSusChem*, 2011, 4(1): 74-78. *Nature*, 2013, 503(7477): 509-512.). Meanwhile, it is found that perovskite generates localized light and heat by electron delay^{10,11} and fluorescence (*J. Phys. Chem. Lett.* 2016, 7(2): 266-271. *Nano Lett.* 2015, 15(7): 4644-4649.), which also can be absorbed by perovskite-encapsulated interface layer. It is extremely sorry that our statement about perovskite is not accurate enough. To ensure rigorous, we changed these unreasonable expressions.

As known, Cs₄PbBr₆ typed perovskite is indeed sensitive to water. When it comes into contact with trace amounts of water, part of the Cs₄PbBr₆ typed perovskite will be converted into CsPbBr₃^{12, 13}. Hence, following the reviewer's suggestion, we directly replaced the Cs₄PbBr₆ type perovskite in the IENG with the CsPbBr₃ type perovskite. X-ray radiation results showed the change of the peak changes of the IENG exposed to water under different evaporation time (Fig. 1h), which confirms the stability of CsPbBr₃ type perovskite in the IENG.

Additionally, as suggested by the reviewers, we demonstrated this claim by comparing the temperature of the IENG with or without perovskite particles under light in the supplementary materials. In Supplementary Fig. 6, we compared the temperature change of the bottom layer without and with perovskite under a solar light intensity of 1.0 kW m⁻² within 1 h. It showed that the surface temperature of the bottom layer with perovskite is higher than that of the bottom layer without perovskite (Supplementary Fig. 6c and Fig. R1). This is attributed to the introduction of the perovskite that improves the light absorption efficiency (Supplementary Fig. 6b).

Fig. 1h. X-ray diffraction pattern of CsPbBr₃ under different evaporation time.

Figure R1. Surface temperature infrared photos of bottom layer with and without perovskite.

Supplementary Figure 6. **b** Light absorption efficiency of the bottom layer with and without perovskite at the wavelength range of 190 nm-900 nm. **c** Surface temperature curves of the bottom layer with and without perovskite under a solar light intensity of $1.0 \text{ kW}\cdot\text{m}^{-2}$.

3. The authors claimed that the TiO₂ and SiO₂ particles absorbed light energy to generate electron-hole pairs. In this case, this part of energy is wasted. Even the energy is released after recombination of electrons and holes, this part energy is dissipated in the hydrogel matrix and not useful to heat the interfacial evaporating layer.

Response: Thanks so much for your constructive suggestion. TiO₂ can absorb UV part of the sunlight to some certain extent (*Catal. Commun.* 2008, 9(6): 1162-1166. *Appl. Surf. Sci.* 2016, 377: 221-227. *ACS Appl. Mater. Inter.* 2018, 10(41): 35316-35326), the

corresponding energy is released after recombination of electrons and holes and can be used to heat the up-flowing water and contribute to a higher water supply to improve interfacial evaporation. However, compared to moisture supply originated from water evaporation by sunlight (J. Mech. Eng. Sci. 227(12): 2665-2670 (2013); Matter, 2(2): 390-403 (2019)), this enhancement effect of light conversion for heating is very weak. Therefore, to ensure rigorous, the components of TiO₂ and SiO₂ of the IENG are removed, and the composition of our IENG is further simplified.

4. What's the exact role of ionic liquid in this hybrid material? How can the concentration of BMIM⁺ ions affect the solar vapor and power generation properties? Could you use other ions to replace the BMIM⁺ ions?

Response: Thanks so much for your constructive suggestion. In this work, ionic liquid of BMIMCl is mainly used to dissolve cellulose to form the hydrogel backbone of the IENG^{14, 15}. After cellulose-BMIMCl is phase separated in deionized water, most of the ions are replaced and the remaining ions very small. The remaining ions (BMIM⁺) can combine with water in the cellulose to form hydrated ions¹⁶. Therefore, water absorption capacity of the IENG is improved to some extent (Supplementary Fig. 7b), which is beneficial to the increase of water evaporation rate of the IENG.

There are many types of ionic liquids that can dissolve cellulose, and BMIMCl can be replaced by other types of ionic liquids. Compared with other types of ionic liquids, EMIMCl and BMIMCl have the highest solubility for cellulose¹⁷, and the mechanical properties of hydrogel formed by dissolving cellulose are also relatively excellent¹⁸. Although the solubility of EMIMCl for cellulose is better than that of BMIMCl, its cost is much higher than that of BMIMCl. In addition, BMIMCl also has a very high recovery rate of 99%¹⁹. Therefore, BMIMCl is the best choice for dissolving cellulose. Considering the dissolving effect and cost-effectiveness, ionic liquid of BMIMCl is selected here.

5. According to my understanding, the power generation in this evaporator is caused by the porous MWNT layer, which acts as an osmotic power generator. Thus the properties of this MWNT layer is critical important in this material. However, the authors didn't provide any characterizations on this part. What's the thickness? What's the porosity? What are the surface properties? How is the charge distribution on surface?

Response: Thanks so much for your constructive suggestion. First, it is necessary to calcify that the working principle of the IENG is traced back to streaming potential theories by pressure-driven flow in nanochannels²¹⁻²³ and recently developed hydrovoltaic effect by fluid flow through narrow solid channels^{2, 24-26}, as shown in *Response to (1)*. Surely, the MWNT layer is extremely important, and we supplemented its characterization analysis as follows.

In Fig. R2a-c, it can be seen that the MWNT layer (called top layer) of the IENG is attached to the surface of the middle layer with a thin (190 μm) grid-like porous structure (porosity about 84.4%). Meanwhile, the MWNT layer is hydrophilic (contact angle 34.33 $^\circ$, see Fig. R2d). Fig. R2e, f showed that there is a small amount of negative charge (-0.43 nC cm^{-2}) on the surface in the dry state. When the surface is in a wet state, the negative charge distribution is greatly increased (-14.2 nC cm^{-2}). After testing, the zeta potential of the IENG was -27.04 mV (Fig. R3).

Fig. R2. Characteristics of the top layer. a, b SEM images of the middle layer and the top layer, c The porosity of the MWNT-microchannel top layer, d Hydrophilic properties of the middle layer and the top layer, e Surface charge distribution of the dried top layer, f Surface charge distribution of the wet top layer.

Fig. R3. Zeta potential curve of the IENG.

6. The osmotic power generation is affected by the ion concentrations. The authors also found that the power generation properties were changed in saline water. So how does the ion concentration of saline water affect the power generation in this material? Will Na^+ or Cl^- ions accumulate in the MWNT layer?

Response: Thanks so much for your constructive suggestion. We carried the power generation experiments under different ion concentrations of saline water. The results showed that as ion concentrations of saline water increases, the power generation performance of the IENG increase as well, further increasing the concentration will lead to a decay of the power generation performance. An optimized power generation performance is presented in sea water (Fig. R4). That is because, a higher concentration means more ions will be selectively dragged through the MWNT layer contributing to a higher power generation. However, as the ion concentration increases, Debye length will significantly reduce leading to the degeneration of ionic selectivity for the MWNT layer. The power generation performance of the IENG will significantly decline (*ACS Appl. Mater. Inter.* 12(9): 11232-11239 (2020). *Angew. Chem. Int. Edit.* 132(26): 10706-10712 (2020)).

Indeed, ion accumulation occurred in the MWNT layer, but no salt crystals appeared in super high ions concentration (Fig. R5). Exactly, the enhanced BL surface with different porosity including BL layer (porosity about 93.19%) and the MWNT layer (porosity about 84.4%) provided a possible way to achieve a dynamic balance of ion concentration to prevent crystal nucleation (*Adv. Mater.* 31, e1900498 (2019). *Energy Environ. Sci.* 11, 1510-1519 (2018).). According to Fick's law, the porosity and the hydrophilic property of the MWNT layer may also provide self-cleaning capacity for surface salt crystallization, and this needs further verification (*Nano Energy* 65,104002

(2019). *J. Membr. Sci.* 586, 222–230 (2019). *ACS Appl. Mater. Inter.* 12, 35142–35151 (2020)). Supplementary experimental results showed that there is accumulation of ions in the MWNT layer under low-concentration saline water, but no salt crystal deposited on the surface. The accumulation of ions in the MWNT layer under high-concentration saline water is relatively serious. This is consistent with the above analysis results.

Fig. R4. Power generation performance of the IENG under different salinity. **a** Residual amount of Na after evaporation of the IENG under different salinity, **b** Open-circuit voltage generated by the IENG under different salinity, **c** Variation of open-circuit voltage produced by the IENG under different salinities, **d** Short-circuit current generated by the IENG under different salinity. **e** Variation of short-circuit current produced by the IENG under different salinity, **f** Power density produced by the IENG under different salinity.

Fig. R5. EDS and SEM images of the IENG in different salinity environment. **a, e** contrast sample. **b, f** samples soaking in sea water for 72 h. **c, g** samples evaporating for 72 h. **d, h** samples evaporating for more than 300 h.

7. *The authors claimed that the enthalpy of water evaporation was changed in this evaporator. What's the enthalpy change? Without this information, how can you calculate the energy conversion efficiency?*

Response: Thanks so much for your constructive suggestion. As reported in the literatures^{16, 17, 20}, the polymer networks of hydrogels can be architected to tune the water state and, hence, to further reduce the evaporation enthalpy of water. Here, we tested vaporization enthalpy of the IENG by differential scanning calorimeter, and it showed the vaporization enthalpy of the IENG is 1111 J g^{-1} (our work) in Fig. R6, while the vaporization enthalpy of water is approximately 2500 J g^{-1} (*Accounts Chem. Res.* **52**(11): 3244-3253 (2019)). We recalculated the energy conversion efficiency of the IENG in Supplementary Fig. 5b.

Fig. R6. Vaporization enthalpy change of IENG by differential scanning calorimeter (DSC).

8. *What's the relationship between vapor generation rate and power generation? Could you measure these values under different light intensities?*

Response: Thanks so much for your constructive suggestion. In this article, the open-circuit voltage, short-circuit current and output power density of the IENG under different light intensities were provided in Fig. 4a-c. As suggested by the reviewers, we supplemented experiments of water evaporation rate and power generation of the IENG with the change of light intensity, and revealed the relationship between water evaporation rate and power generation under different light intensities. The detailed experimental results are shown in Supplementary Figure 9. The specific analysis is as follows.

When the external conditions remain unchanged, it can be found that by continuously increasing the ambient light intensity of 2 kW m^{-2} , water evaporation rate of the IENG has been greatly increased to $4.385 \text{ kg m}^{-2} \text{ h}^{-1}$ (Supplementary Figure 9a, b). Through the analysis, it can be found that water evaporation rate of the IENG has an exponential function relationship with the light intensity (Supplementary Figure 9c). This is mainly because as the light intensity increases, the energy absorbed by the IENG increases, which in turn increases its water evaporation rate. However, since water evaporation rate of the IENG is also limited by water absorption properties of its materials, and it has a maximum value. Meanwhile, it is also found that the open-circuit voltage and the short-circuit current generated by the IENG increase with the increasing intensity of light received. By testing the open-circuit voltage and the short-circuit current generated by the IENG under different light intensities, it is found that the open-circuit voltage and the short-circuit current generated by the IENG increase with the increase of the light intensity (Supplementary Figure 9d, f). This is because the open-circuit voltage and the short-circuit current generated by the IENG are mainly produced by its water evaporation process. Therefore, the relationship between the open-circuit voltage and the short-circuit current of IENG and the change of light intensity is similar to the relationship between water evaporation rate and the light intensity (Supplementary Figure 9e, g). By testing the maximum output power of the IENG under different light intensities, it can be found that the output power density also increases with the increase of light intensity (Supplementary Figure 9h). The output power density of the IENG also has an exponential relationship with light intensity (Supplementary Figure 9i).

In addition, combining with the open-circuit voltage, the short-circuit current and the power density analysis under different light intensities, a relationship of the IENG between power generation and water evaporation rate under different light intensities is obtained (Supplementary Fig. 9j-l). The power generation of the IENG increases exponentially with the improvement of its water evaporation rate under different light intensities, which is in line with the theoretical formulas obtained in Supplementary Note 3 and Supplementary Note 4 (*ACS Appl. Mater. Inter.* 2020, 12(9): 11232-11239).

Supplementary Figure 9. Relationship of the power generation performance, light intensity and water evaporation rate. **a** Mass loss of the IENG under different light intensities. **b** Evaporation rate of the IENG under different light intensities. **c** Relationship between the evaporation rate and the light intensity. **d** Open-circuit voltage under different light intensities **e** Relationship between the open-circuit voltage and the light intensity. **f** Short-circuit-current under different light intensities. **g** Relationship between the short-circuit current and the light intensity **h** Power density under different light intensities. **i** Relationship between the power density and the light intensity. **j-l** Relationship between power generation and water evaporation rate under different light intensities.

Fig. 4 Power generation performance of the IENG under different surrounding conditions. **a-c** Power generation performance of the IENGs under different light intensities. **d-f** Power generation performance of the IENGs in different liquids. **g-i** Power generation performance of the IENGs under different wind speeds. The lengths, widths and heights were 20 mm, 20 mm and 20 mm, respectively.

9. The authors measured the properties of their material in marine environment under 2 kW/m² light intensity. How can you get such high light intensity in marine environment? The solar intensity on earth surface is usually under 1 kW/m².

Response: Thanks so much for your constructive suggestion. Efficient light-to-heat conversion has always been the pursued goal by researchers for vapor evaporation. The increasing in the light intensity on the surface means to improve the efficiency of light-to-heat conversion. Hence, there are many approaches that can achieve light intensity from 1 kW m⁻² to 2 kW m⁻².

Here, the experimental device designed by us is utilized for fresh water collecting and energy harvesting, as shown in Fig. R7. The upper part of the device is equipped with a Fresnel lens, which can effectively enhance sunlight from 1 kW m⁻² to 2 kW m⁻². Therefore, it is entirely possible to obtain a light intensity of 2 kW m⁻² in a marine environment.

Fig. R7. Illumination enhancement equipment for power generation and freshwater production.

10. Panel g and f in Supplementary Figure 2 are missing.

Response: Thanks so much for your constructive suggestion. We have removed panels of g and f in Supplementary Figure 2 as redundant titles.

11. So many details are missing in the Methods Section. The authors should carefully describe how they conduct the solar vapor generation and power generation measurements.

Response: Thanks so much for your constructive suggestion. We supplemented the missing details in the method section. The details of solar vapor generation and power generation measurements are carefully described below.

Measurement of solar vapor generation. The IENG was placed in an experimental cistern (30 °C, 40% RH, summer, in Harbin, China). A solar vapor evaporation device was placed on an electronic balance and illuminated by solar simulator to monitor evaporation quantity in real time. The mass change of sea water in the solar evaporator was recorded transiently by an electronic balance.

Measurement of power generation. The power generation measurement used the solar vapor evaporation device to supplement wind energy and other modules to simulate marine environment (21.4 °C, 15.8% RH, winter, in Harbin, China). Before the electrical performance test, a stable conductive system was constructed by intermittently dropping polyaniline/ethanol on the surface. The short-circuit current test is concentrated on the depth of the boundary of the middle-top-layer interface, and the open-circuit voltage test is connected to the top layer surface and

approximately the middle height of the middle layer. Here, gold electrodes were selected to test the electricity performance (Supplementary Fig. 16). The upper and lower test positions need to be dynamically adjusted at the first time. The power generation performance of the IENG was measured after connected to a Keithley 6514 electrometer (USA) under different light intensities by solar simulator.

Supplementary References for Response Letter of Reviewer #1

1. Ding, T. et al. All-printed porous carbon film for electricity generation from evaporation-driven water flow. *Adv. Funct. Mater.* **27**, 1700551 (2007).
2. Liu, K., Ding, T. & Li, J. et al. Thermal–electric nanogenerator based on the electrokinetic effect in porous carbon film. *Adv. Funct. Mater.* **8**, 1702481 (2018).
3. Jia, L. et al. Surface functional modification boosts the output of an evaporation-driven water flow nanogenerator. *Nano Energy*. **58**, 797-802 (2019).
4. Kortschot, R. J. et al. Debye length dependence of the anomalous dynamics of ionic double layers in a parallel plate capacitor. *J. Phys. Chem. C* **118**, 11584-11592 (2014).
5. Kohonen, M.M. et al. Debye length in multivalent electrolyte solutions. *Langmuir* **16**, 5749-5753 (2000).
6. Yin, J. et al. Harvesting energy from water flow over graphene? *Nano Lett.* **12**, 1736-1741 (2012).
7. Wang, X. et al. Dynamics for droplet-based electricity generators. *Nano Energy* **80**, 105558 (2021).
8. Van der Heyden, F. H. J., et al. Power generation by pressure-driven transport of ions in nanofluidic channels. *Nano Lett.* **7**, 1022-1025 (2007).
9. Liu, A. et al. Direct electricity generation mediated by molecular interactions with low dimensional carbon materials-A mechanistic perspective. *Adv. Energy Mater.* **8**, 1802212 (2018).
10. Piatkowski, P. et al. Direct monitoring of ultrafast electron and hole dynamics in perovskite solar cells. *Phys. Chem. Chem. Phys.* **17**(22): 14674-14684 (2015).
11. Long, R. Fang, W. Prezhdo, O.V. Moderate humidity delays electron–hole recombination in hybrid organic–inorganic perovskites: Time-domain ab initio simulations rationalize experiments. *J. Phys. Chem. Lett.* **7**(16): 3215-3222 (2016).
12. Wang, X. et al. The effects of hydroxyl by water addition on the photoluminescence of zero-dimensional perovskites Cs₄PbBr₆ nanocrystals. *J. Lumn.* **221**: 116986 (2020).
13. Wu, L. et al. From Nonluminescent Cs₄PbX₆ (X = Cl, Br, I) Nanocrystals to Highly Luminescent CsPbX₃ Nanocrystals: Water-Triggered Transformation through a CsX-Stripping Mechanism. *Nano Lett.* **17**(9): 5799-5804 (2017).
14. Kramer, R. K. & Carvalho, A. J. F. Non-freezing water sorbed on microcrystalline cellulose studied by high-resolution thermogravimetric analysis. *Cellulose*. **28**(16): 10117-10125 (2021).
15. Feng, L. & Chen, Z. I. Research progress on dissolution and functional modification of cellulose in ionic liquids. *J. Mol. Liq.* **142**(1-3): 1-5 (2018).
16. Zhao, D. et al. A dynamic gel with reversible and tunable topological networks and performances. *Matter*, **2**(2): 390-403 (2019).
17. Zhou, X. et al. Hydrogels as an emerging material platform for solar water purification. *Accounts*

- Chem. Res.* **52**(11): 3244-3253 (2019).
18. Zhou, X. et al. Topology-controlled hydration of polymer network in hydrogels for solar-driven wastewater treatment. *Adv. Mater.* **32**(52): 2007012 (2020).
 19. Kim, K. B. & Kim, J. Fabrication and characterization of electro-active cellulose films regenerated by using 1-butyl-3-methylimidazolium chloride ionic liquid. *J. Mech. Eng. Sci.* **227**(12): 2665-2670 (2013).
 20. Shin, S. et al. Three-body hydrogen bond defects contribute significantly to the dielectric properties of the liquid water–vapor interface. *J. Phys. Chem. Lett.* **9**(7): 1649-1654 (2018).
 21. Burgreen, D. & Nakache, F.R. Efficiency of pumping and power generation in ultrafine electrokinetic systems. *J. Appl. Mech.* **32**, 675 (1965).
 22. Yang, J. et al. Electrokinetic microchannel battery by means of electrokinetic and microfluidic phenomena. *J. Micromech. Microeng.* **13**(6): 963 (2003).
 23. Zuo, G. et al. Transport properties of single-file water molecules inside a carbon nanotube biomimicking water channel. *Acs Nano*, **4**(1): 205-210 (2010).
 24. Yin, J. et al. Generating electricity by moving a droplet of ionic liquid along graphene. *Nat. Nanotechnol.* **9**, 378-383 (2014).
 25. Ding, T. et al. All-printed porous carbon film for electricity generation from evaporation-driven water flow. *Adv. Funct. Mater.* **27**(22): 1700551 (2007).
 26. Xue, G. et al. Water-evaporation-induced electricity with nanostructured carbon materials. *Nat. Nanotechnol.* **12**, 317-321 (2017).

Reviewer #2 (Remarks to the Author):

Solar powered interfacial evaporators have a great potential in delivering clean water and electric power simultaneously from a single device. Such hybrid devices would be valuable everywhere from crowded cities to remote areas where the use of land is limited, or electricity and clean water are not readily available. New technologies are needed to harness both forms of energy as efficiently as possible, ideally at low cost. Sun and co-workers have developed a 3D structured interfacial evaporator mimicking the natural surface structure of moth's eye. The device not only delivered a respectful evaporation rate, but also had a secondary function of driving a nanogenerator, meaning that the evaporation process itself generated direct current. In my view, the biggest value of this work comes from the nature-inspired moth's eye design that improved light trapping and thereby the overall efficiency of the hybrid device. This intriguing system is presented in an interesting way considering the readership of the journal, making a significant impact in the field. This is a solid piece of work with comprehensive characterization and data analysis. However, this work has potential for even better in terms of readability and rigorous grammatic revision to meet the standards of Nature Communications. Some general comments to the authors:

Response: Thank you very much for your recognition and recommendation of our work. According to the requirements, we have carefully revised the language and statement to make our work more rigorous and readable. Multiple language polish was performed by Springer Nature Authors Services (Language Paper ID: FE37-01B7-7B75-0606-7ED4 & E5F6-59BC-3609-D23E-96CC & F778-F8A9-0A1F-8182-A94B) to eliminate grammatic errors.

Meanwhile, in response to the reviewers' comments, we conducted an extensive literature search. In this process, we found that experimental error may be caused by electrochemical corrosion or polarization of test electrodes (*Nat. Nano.* 12(4): 317-321 (2017). *Angew. Chem. Int. Edit.* 132(26): 10706- 10712. (2020). *ACS Appl. Mater. Inter.* 12(9): 11232-11239 (2020)). In order to make our experimental results more rigorous and comparable, we replaced the original copper (Cu) wrapped with tin (Sn) electrodes (~1.1 V, ~220 μA , 35.14 $\mu\text{W cm}^{-2}$) with gold (Au) electrodes (543 mV, ~121.4 μA , 11.8 $\mu\text{W cm}^{-2}$), and re-executed the power generation experiment with Au test electrodes in

Fig. 2, Fig. 4, and supplementary materials. Here, it should be explained that the difference in power generation performance produced by test electrode materials will be further studied in our follow-up work. It is found that the open circuit voltage per unit area with Au electrodes is improved by a factor of 9.82 over the currently reported average value (*Nano Energy* 57, 269-278 (2019), Supplementary Fig. 4a).

Additionally, the power density is more objectively to evaluate the power generation performance of the device compared to the previously used open circuit voltage per unit area. Thus, we found that a breakthrough result of the power density with Au electrodes is $11.8 \mu\text{W cm}^{-2}$ under a light intensity of $2 \text{ kW}\cdot\text{m}^{-2}$, which is improved by a factor of 6.8 over the currently reported average value (Fig. 5d).

We believe that these above results are more objectively to reflect the performance and advantages of our IENG. Therefore, we revised the full manuscript according to the reviewer's suggestion and combined with the above ideas. Thank you so much for your understanding.

1) The title in the SI file is different from the main manuscript. Please correct.

Response: Thanks so much for your constructive suggestion. The title in the SI file has been revised.

Achieving efficient power generation by designing bioinspired and multi-layered interfacial evaporator

2) It is not clear to me what the authors want to say with the phrase “greatly potential for sustainable spontaneously power generation” in the abstract and later in the text. Could you please clarify? Similarly, some sentences in the abstract are rather long and hard to follow, such as “Simultaneously, inspired by the light-trapping properties of moth eye, a simple and efficient IENG including a hierarchical surface of bionic light-trapping and electrospinning perovskite conductivity with an enhanced thermally insulating and water storage capability is designed.” Also, “This work provides an unexplored strategy for multi-energies inspired natural interfacial evaporation driven power generation.” I would recommend splitting the sentences shorter and revising them so that they read better.

Response: Thanks so much for your constructive suggestion. As recommended by the reviewers, we have proceeding the splitting the sentences shorter and revising to make

our article readable. The specific modifications of the statements are as follows.

(a) The phrase “greatly potential for sustainable spontaneously power generation” in the abstract and later in the text has been revised.

(b) “Simultaneously, inspired by the light-trapping properties of moth eye, a simple and efficient IENG including a hierarchical surface of bionic light-trapping and electrospinning perovskite conductivity with an enhanced thermally insulating and water storage capability is designed.” This sentence in the abstract has been revised to read clearly.

(c) “This work provides an unexplored strategy for multi-energies inspired natural interfacial evaporation driven power generation.” This sentence also has been revised.

***Abstract:** Water evaporation is a natural phase change phenomenon occurring any time and everywhere. Enormous efforts have been made to harvest energy from this ubiquitous process by leveraging on the interaction between water and materials with tailored structural, chemical and thermal properties. Here, we develop a multi-layered interfacial evaporation-driven nanogenerator (IENG) that further amplifies the interaction by introducing additional bionic light-trapping structure for efficient light to heat and electric generation on the top and middle of the device. Notable, we also rationally design the bottom layer for sufficient water transport and storage. We demonstrate the IENG performs a spectacular continuous power output as high as $11.8 \mu\text{W cm}^{-2}$ under optimal conditions, more than 6.8 times higher than the currently reported average value. We hope this work can provide a new bionic strategy using multiple natural energy sources for effective power generation.*

3) At the end of page 2, the authors write “...large thermal insulating capacity (increased by 1.5 times)”. Please specify what is that 1.5 times increase compared to. On pages 3–4, please describe what are “control 1” and “control 2”, where first mentioned.

Response: Thanks so much for your constructive suggestion. Here, we have completed the amendment to the above statement, and the specific amendments are as follows.

(a) 1.5 times increase in the sentence “...large thermal insulating capacity (increased by 1.5 times)” is compared to ITW-layer without the thermal insulation. This

statement has been deleted in the article on page 2.

- (b) We modified this expression of “control 1” and “control 2”. And “control 1” and “control 2” were changed as “ionic hydrogel” and “ionic hydrogel with MWNTs” in the article on page 3 and Fig. 2a, b.

Fig. 2 Power generation performance and evaporation characteristics of the IENG. a Open-circuit voltage per unit area of the IH, IH with MWNTs and IENG. **b** Short-circuit currents per unit area for the IH, the IH with MWNTs and the IENG.

- 4) The text in Fig. 5b,c is hard to read. Please increase the text size.

Response: Thank you so much for your constructive suggestions. We have made corrections to the descriptions in Figures 5b and captions so that readers can see and understand this part of the data more clearly. Considering that the power density is more objectively to evaluate the power generation performance of the device compared to the previously used open circuit voltage per unit area, the power density was selected for comparison. As needed, we changed the power density of the IENG in Fig. 5c to Fig. 5d.

Fig. 5 Working principle and application of the IENG. b Water evaporation rate of the IENG in a simulated marine environment. The normal environment had a $1.0 \text{ kW} \cdot \text{m}^{-2}$ light intensity, no wind and deionized water, and the optimized environment had a $2.0 \text{ kW} \cdot \text{m}^{-2}$ light intensity, a 1 m s^{-1} wind speed and sea water. **c** Output power density of the IENG measured under different load resistances. **d** Comparison of power density generated by the IENG and other solar-driven IENGs.

5) *Related to one of the comments above, I would suggest shortening and clarifying some of the sentences in the conclusion as well. Such as “In summary, we demonstrated a solar and wind natural energies driven interfacial evaporation nanogenerator (IENG) including a hierarchical surface of bionic light-trapping and electrospinning perovskite conductivity with an enhanced thermally insulating and water storage capability.”*

Response: Thanks a lot for your constructive suggestion. As suggested, some of the sentences in the conclusion is also shortened and clarified, which is highlighted in blue.

Conclusion: *In summary, we have demonstrated an efficient multi-layered interfacial evaporation-driven power generation system mimicking the natural structure of the moth's eye surface. The advantages of this device include excellent moisture storage/ supply ability, outstanding solar-heat-conversion property and remarkable electric conductivity, which allows it to efficiently utilize the ambient low-grade heat for freshwater and power generation. Under the modified condition, our device performs an excellent freshwater production of $2.78 \text{ kg m}^{-2} \text{ h}^{-1}$ with a light intensity of $1.0 \text{ kW}\cdot\text{m}^{-2}$ and an electric output power density of $11.8 \mu\text{W cm}^{-2}$, which is over 6.8 times larger than the average devices. The synergistic effect from enhanced evaporation and hydrovoltaic effect contributes to such a good performance. Therefore, this work demonstrates a sustainable interfacial evaporation-driven power generation approach and provides a foundation for utilizing multiple natural energy sources. It can serve to develop offshore power generation platforms and freshwater supply devices in the future.*

6 *It would very be attractive to see discussion about the scalability and cost-effectiveness of the moth's eye design and the chosen perovskite, carbon nanotube, etc. materials. Surely, the cost of the devices plays another important role when thinking about their potential use. Is there anything to develop further in that regard? At least as “future perspectives” in the conclusion section. How would the water evaporation rate and power output scale in realistic conditions?*

Response: Thanks so much for your constructive suggestion. The scalability and cost-effectiveness of the moth's eye design of the IENG are indeed critically significant

issues to be considered in the future. Water production and power generation integrated device has broad promotion potential in the future (Fig. R7). The end section of this article supplemented discussion of future developments of this technology, and revealed the relationship between water evaporation rate and power generation under different light intensities in Supplementary Fig. 9. The specific parameters of water evaporation rate and power generation in realistic conditions are given in the article.

In the future, this IENG with a bionic moth eye structure for a spontaneous evaporation-driven power generation strategy has realistic possibility and scalability. From technical perspectives, our IENG device only needs to be etched a biological template and then sprayed with perovskite and carbon nanotubes, etc. It is relatively simple in operation and convenient in processing. In addition to the expensive material cost of perovskites and carbon nanotubes, the cost of other materials is relatively cheaper. After statistical analysis, our IENG device only costs about 800 RMB m^{-2} . From the perspective of power generation, experiments showed that the IENG can collect about $2.19 \text{ kg h}^{-1} m^{-3}$ of clean freshwater, and produce about 10 W h m^{-3} of electricity energy. Therefore, this device can be widely deployed on sea for freshwater production and electric energy harvesting, which creates an opportunity to develop offshore power generation platforms and freshwater supply devices.

The detailed analysis is supplemented on Page 7 of this article.

We also did a statistical analysis. With a freshwater production of $2.19 \text{ kg h}^{-1} m^{-3}$ and an electrical generation of 10 W h m^{-3} , our IENG device only costs approximately 800 RMB m^{-2} showing its potential in realisticity and scalability. Furthermore, we designed a self-powered electronic integrated system. As shown in Fig. 5e, a low-voltage device can operate continuously with only solar and wind energy input, demonstrating an opportunity to develop offshore power generation platforms and fresh water supply devices (working system circuit in Supplementary Fig. 15a b).

Fig. R7. Illumination enhancement equipment for power generation and freshwater production.

Supplementary Figure 9. Relationship of the power generation performance, light intensity and water evaporation rate. **a** Mass loss of the IENG under different light intensities. **b** Evaporation rate of the IENG under different light intensities. **c** Relationship between the evaporation rate and the light intensity. **d** Open-circuit voltage under different light intensities **e** Relationship between the open-circuit voltage and the light intensity. **f** Short-circuit-current under different light intensities. **g** Relationship between the short-circuit current and the light intensity **h** Power density under different light intensities. **i** Relationship between the power density and the light intensity. **j-l** Relationship between power generation and water evaporation rate under different light intensities.

REVIEWERS' COMMENTS

Reviewer #1 (Remarks to the Author):

The authors have addressed my concerns. I have no further questions.

Reviewer #2 (Remarks to the Author):

The authors have rigorously revised the manuscript and addressed all my previous comments. The language of the revised manuscript has been greatly enhanced and the manuscript is significantly more pleasing to read in its revised form. To me the experimental work is well described considering reproducibility and the results have been clearly and concisely discussed in the manuscript.

Considering all these improvements, I have no further comments and I can recommend the revised manuscript to be published in Nature Communications.